# Granulocyte Colony Stimulating Factor Expression in Breast Cancer and Its Association with Carbonic Anhydrase IX and Immune Checkpoints

**DOI:** 10.3390/cancers13051022

**Published:** 2021-03-01

**Authors:** Shawn C. Chafe, Nazia Riaz, Samantha Burugu, Dongxia Gao, Samuel C. Y. Leung, Anna F. Lee, Cheng-Han Lee, Shoukat Dedhar, Torsten O. Nielsen

**Affiliations:** 1Department of Integrative Oncology, BC Cancer Research Centre, Vancouver, BC V5Z 1L3, Canada; schafe@bccrc.ca (S.C.C.); sdedhar@bccrc.ca (S.D.); 2Department of Pathology and Laboratory Medicine, University of British Columbia, Vancouver, BC V6H 3Z6, Canada; nazia.riaz@ubc.ca (N.R.); samanthabur@gmail.com (S.B.); dongxia.gao@vch.ca (D.G.); samuel.leung@vch.ca (S.C.Y.L.); anna.lee@cw.bc.ca (A.F.L.); chenghan.lee@bccancer.bc.ca (C.-H.L.); 3Centre for Regenerative Medicine and Stem Cell Research, Aga Khan University, Sindh 74800, Pakistan; 4Department of Biochemistry and Molecular Biology, University of British Columbia, Vancouver, BC V6T 1Z3, Canada

**Keywords:** granulocyte colony-stimulating factor, carbonic anhydrase IX, hypoxia, tumour-associated macrophages, tumour-infiltrating lymphocytes, invasive breast cancer, prognosis

## Abstract

**Simple Summary:**

Preclinical studies suggest that interactions between granulocyte colony-stimulating factor (G-CSF) and hypoxia-induced carbonic anhydrase IX regulate the trafficking and function of immune cells in the tumour microenvironment. We investigated the clinical significance of this crosstalk by analyzing the protein expression of G-CSF and macrophage markers by immunohistochemistry on a well-characterized tissue microarray series of invasive breast cancers. We report that high expression of G-CSF on breast carcinoma cells is linked with significantly improved survival in an important group of breast cancers that do not respond to hormonal therapy. These tumours were infiltrated by immune cells expressing biomarkers that can be targeted with immune checkpoint inhibitor drugs. In contrast, carbonic anhydrase IX expression was associated with unfavourable outcomes.

**Abstract:**

Purpose: Granulocyte colony-stimulating factor (G-CSF) and hypoxia modulate the tumour immune microenvironment. In model systems, hypoxia-induced carbonic anhydrase IX (CAIX) has been associated with G-CSF and immune responses, including M2 polarization of macrophages. We investigated whether these associations exist in human breast cancer specimens, their relation to breast cancer subtypes, and clinical outcome. Methods: Using validated protocols and prespecified scoring methodology, G-CSF expression on carcinoma cells and CD163 expression on tumour-associated macrophages were assayed by immunohistochemistry and applied to a tissue microarray series of 2960 primary excision specimens linked to clinicopathologic, biomarker, and outcome data. Results: G-CSF_high_ expression showed a significant positive association with ER negativity, HER2 positivity, presence of CD163+ M2 macrophages, and CAIX expression. In univariate analysis, G-CSF_high_ phenotype was associated with improved survival in non-luminal cases, although the CAIX+ subset had a significantly adverse prognosis. A significant positive association was observed between immune checkpoint biomarkers on tumour-infiltrating lymphocytes and both G-CSF- and CAIX-expressing carcinoma cells. Immune checkpoint biomarkers correlated significantly with favourable prognosis in G-CSF_high_/non-luminal cases independent of standard clinicopathological features. Conclusions: The prognostic associations linking G-CSF to immune biomarkers and CAIX strongly support their immunomodulatory roles in the tumour microenvironment.

## 1. Introduction

Granulocyte colony-stimulating factor (G-CSF) is a member of the colony-stimulating factor superfamily and exerts its biological effects by binding to the G-CSF receptor (G-CSFR). The archetypal G-CSF/G-CSFR signalling pathway is critical for proliferation and survival of myeloid precursors and their subsequent differentiation into neutrophils with augmentation of their effector functions [1]. Recognition of these biological functions led to the establishment of G-CSF as a prophylactic and therapeutic agent for chemotherapy-induced febrile neutropenia [2]. Considering the robust relationship between inflammation and cancer, the role of G-CSF- and G-CSF-mobilized immune cells has been investigated in preclinical models of several non-hematopoietic malignancies, including breast cancer [3,4]. These studies have implicated aberrant G-CSF/G-CSFR signalling with altered hematopoiesis, leading to the recruitment of immunosuppressive cells in the tumour microenvironment that potentiate migration, invasion, angiogenesis and metastasis [5,6,7,8,9,10,11,12,13]. In line with preclinical data, clinical studies evaluating G-CSF expression on tumour cells have also shown an association with aggressive clinicopathological features and poor prognosis in solid organ malignancies such as clear cell renal cell carcinomas and cervical cancers [14,15].

Tumour hypoxia is a hallmark of invasive tumours and is associated with genomic instability, the emergence of therapy-resistant clones, inhibition of anti-tumour immune responses, and cancer progression [16,17]. Carbonic anhydrase IX (CAIX) is a cell surface membrane-bound enzyme expressed by hypoxic tumours. CAIX catalyzes the reversible hydration of CO_2_ into bicarbonate ions and protons. The bicarbonate is transported into the cells via bicarbonate transporters, thereby aiding in the buffering of intracellular pH, whereas the protons remain in the extracellular space, contributing to extracellular acidification [18]. We and others have previously shown that immunohistochemical expression of CAIX is frequently associated with features of aggressive disease and adverse survival in breast cancer [19,20,21]. In preclinical models of breast cancer, G-CSF production by the hypoxic tumour cells, in a CAIX-dependent manner, facilitated the formation of a lung premetastatic niche by recruiting myeloid-derived suppressor cells [22]. Furthermore, in the same model, CAIX inhibition enhanced the efficacy of immune checkpoint blockade [23]. Thus, there is strong evidence to support that the hypoxic, acidic microenvironment of the tumours is immunosuppressive and enriched in a range of cells, including myeloid-derived suppressor cells, T lymphocytes, and tumour-associated macrophages (TAMs), all of which contribute to immune tolerance by dampening anti-tumour effector cell functions [16,24,25,26].

Tumour-associated macrophages comprise heterogeneous populations of cells that exhibit remarkable plasticity and play an important role in modulating adaptive immune responses [27]. However, their biological role and prognostic significance are dependent upon phenotype and localization, which can be assessed using biomarkers. The classically activated M1 macrophages are characterized by the production of proinflammatory cytokines, enhanced antigen presentation, tumouricidal effects and improved cancer outcomes [28,29]. In contrast, alternatively activated M2 macrophages are associated with anti-inflammatory cytokines, immunosuppressive responses and protumoural properties. M2 macrophages are reliably identified by the expression of CD163, which is a cell surface glycoprotein and a member of the scavenger receptor superfamily class B [27]. Tumour infiltration with CD163+ M2 macrophages is associated with features of aggressive disease and poor outcomes in breast cancer patients, notably those diagnosed with HER2 and basal subtypes [30,31]. Preclinical studies have shown that G-CSF secretion by triple-negative breast cancer cell lines potentiates macrophage differentiation into an immunosuppressive phenotype associated with enhanced migratory capacity [32]. In addition, it has been shown that hypoxic tumour cells also promote M2-like polarization of tumour-infiltrating macrophages [33].

Thus, existing data from preclinical studies highlight hypoxia and inflammation as critical modulators of the immune microenvironment of solid tumours where a significant interplay between G-CSF and CAIX has been found to play a role. Presently, it is not known whether there is an association amongst G-CSF, CAIX, and immune biomarkers in breast cancer clinical samples. Furthermore, to date, only limited studies (with insufficient statistical power) have evaluated the prognostic significance of G-CSF expression in breast cancer [32]. Hence, we hypothesized that high expression of G-CSF would correlate with features associated with breast cancer aggressiveness, including CAIX expression, and provide prognostic information across breast cancer subtypes. The objectives of this study were to investigate the immunohistochemical expression of G-CSF in a large series of breast cancers powered for correlation with clinicopathological features, CAIX expression, and immune checkpoint biomarkers; and to examine the prognostic significance of G-CSF in relation to CAIX and CD163+ M2 macrophages using breast cancer-specific survival as the primary endpoint, and overall survival and relapse-free survival as secondary endpoints.

## 2. Materials and Methods

### 2.1. Study Cohorts

We first established the immunohistochemical (IHC) staining procedure, scoring methodology, and interpretation of G-CSF protein expression on a series of female breast cancer patients diagnosed with stage I–III disease during 1998–2002 (cohort I: *n* = 330). The clinicopathological characteristics, adjuvant treatments, and inclusion/exclusion criteria have been described previously [34]. The median follow-up of this cohort was 13 years. We conducted validation and exploratory analyses on an independent, much larger series (cohort II: *n* = 2960) of female patients diagnosed with stage I–III breast cancer in the province of British Columbia at the British Columbia Cancer Agency between 1986–1992, for which formalin-fixed paraffin-embedded tumour blocks were accessible. The adjuvant systemic and endocrine therapies recommended in the specified years were in accordance with the provincial standards. No patients received immune checkpoint inhibitor therapy. The clinicopathological characteristics of this cohort have been formerly described [35,36]. All cases diagnosed with ductal carcinoma in-situ only, stage IV disease at presentation, and those receiving neoadjuvant chemotherapy regimens were excluded from this study.

The Clinical Research Ethics Board of the University of British Columbia and the Breast Cancer Outcomes Unit of BC Cancer approved access to the clinical samples and to corresponding deidentified clinical data. The study was conducted in accordance with the Reporting Recommendations for Tumor Marker Prognostic Studies (REMARK) guidelines [37].

### 2.2. Tissue Microarrays and Immunohistochemistry

The construction of the tissue microarrays used in this study has been described previously [34,38,39]. In all, there were 20 tissue microarray blocks (0.6 mm cores) built from paraffin-embedded primary surgical specimens. Of these, 3 blocks represented cohort I (2 cores per case), and 17 were built from cohort II (1 core per case). Other biomarkers included in studies of cohort II (ER, PR, HER2, Ki-67, CK5/6, EGFR, CD8, FOXP3, LAG3, TIM3, PD-L1, PD-1, CAIX) and IHC-based intrinsic breast cancer subtyping have been described in previous publications [19,34,36,40,41,42,43]. The assessment criteria for stromal tumour-infiltrating lymphocytes (sTILs) on hematoxylin- and eosin-stained sections was in accordance with the recommendations of the International TIL Working Group [44] and the cases were categorized into two groups using <10% vs. ≥10% score as described previously [45].

The IHC protocols are detailed in Appendix A. The stained slides were digitally scanned using a BLISS system (Bacus Laboratories/Olympus America, Lombard, IL, USA) and visually scored by the pathologists (DG, AFL, CHL) who were blinded to the clinical characteristics and outcome data.

The scoring of G-CSF expression was adapted from Hollmen et al. [32]. The intensity of cytoplasmic expression in breast carcinoma cells was classified into four categories: score 0, no reactivity; score 1, weak cytoplasmic reactivity; score 2, moderate cytoplasmic reactivity; score 3, strong cytoplasmic reactivity. For cohort I, the mean intensity score of the duplicate cores was estimated. Cases scored as ≤ 1 were considered low expressors of G-CSF, >1 high. For CD163 and CAIX, we used previously published criteria as follows. Membranous or cytoplasmic expression of CD163 was scored on tumour-associated macrophages as previously described [46] and was classified into three categories: score 1 (sparse infiltrates, ≤5 stained macrophages); score 2 (moderate infiltrates, >5 but ≤25 positively stained macrophages); and score 3 (dense infiltrates, >25 positively stained macrophages). For CAIX, membranous expression on carcinoma cells was scored as 1 (any reactivity) and 0 (no reactivity), as described previously [19]. Cores with insufficient diagnostic material or with equivocal staining results were omitted from further analysis. Images from these slides are available for public access via the website of the Genetic Pathology Evaluation Center (http://www.gpec.ubc.ca/gcsf accessed on 1 January 2021), and representative images are shown in Figure 1. Representative images from serial section of a single core are shown in Appendix A.

### 2.3. Statistical Analysis

IBM^®^ SPSS software (version 25) and R (v 3.3.2) were used for statistical analyses. Relevant descriptive statistics were computed for continuous and categorical variables. The correlation of G-CSF with clinicopathological factors and key biomarkers was assessed by chi-square. The primary endpoint for the clinical outcome was breast cancer-specific survival (BCSS), specified as the time interval between the date of diagnosis and the date of breast cancer-associated mortality. Patients who were alive at the end of follow-up or died due to non-breast cancer-related causes were censored. Overall survival (OS) and relapse-free survival (RFS) were used as secondary endpoints. OS was defined as the time period from the date of diagnosis till the date of last follow-up or death due to any cause, whereas RFS was defined as the time period from the date of diagnosis until any breast cancer-related relapse, whether local, regional, distant, or contralateral. Univariate analysis for survival probabilities was computed by the Kaplan–Meier method, and differences in the survival rates between analyzed groups were estimated by log-rank test. Cox proportional hazard modelling was used for multivariable analysis, and adjusted hazard ratios with 95% confidence intervals were reported for each variable included in the model. A *p*-value of <0.05 was considered statistically significant. To address the concerns of multiple comparisons while assessing associations between G-CSF expression and clinicopathological features, a Bonferroni correction was applied, making the criterion for statistical significance *p* < 0.003.

## 3. Results

### 3.1. Correlation of G-CSF Expression with Clinicopathological Features and Survival

Cohort I was used to evaluate the performance of G-CSF immunostaining, finalize the scoring methodology, and explore cut points such that these could be locked down prior to assessment of the main study cohort II. The correlative analyses with clinicopathological features and association with primary and secondary endpoints for cohort I are presented in Appendix A). On this smaller series, no significant clinicopathologic associations were observed. On multivariable analysis, after adjusting for standard clinicopathological features, high expression of G-CSF was significantly associated with better prognosis (Appendix A).

We describe hereafter the results of our detailed analysis on cohort II (*n* = 2960), data from which has been previously published for expression of immune biomarkers and CAIX [19,40,42,43].

For cohort II, the mean age of the patients at the time of diagnosis was 58.9 years (range: 23–95 years), and the median duration of follow-up was 12.5 years (range: 0.1–18.5 years). A total of 1956 deaths were recorded in the entire cohort, of which 58.4% were attributed to breast cancer. The clinicopathological characteristics of the study cohort are summarized in Table 1.

Of the 2960 cores, high expression of G-CSF was observed in 46.7% of tumours. After correcting for multiple comparisons, these cases demonstrated significantly higher rates of estrogen receptor (ER) negativity, HER2 positivity, CD163+ M2 macrophages, CAIX expression, and (IHC-defined) HER2 and basal intrinsic breast cancer subtypes, relative to the expression of these biomarkers among the G-CSF_low_ cases that comprised 53.3% of the study population (Table 1).

In univariate analysis, G-CSF expression was not prognostically informative on the whole cohort (HR 0.95, CI 0.83–1.08; *p* = 0.43) nor when the analysis was restricted to the luminal subtype, which comprised 74% of the G-CSF interpretable cases (HR 0.99, CI 0.84–1.16; *p* = 0.90). In contrast, amongst the non-luminal subtype, high expression of G-CSF correlated with a significantly improved BCSS (HR 0.74, CI 0.58–0.95; *p* = 0.02) (Figure 2). Similarly, high G-CSF was also associated with significantly improved OS and a trend toward a better RFS in non-luminal cases (Appendix A).

Consistent with the univariate analysis, G-CSF expression was not an independent prognostic indicator in the whole cohort (*n* = 2960) using Cox proportional hazards model for multivariate analysis (HR 0.92, CI 0.80–1.06; *p* = 0.26). However, within the non-luminal cases (*n* = 665), G-CSF_high_ tumours correlated with a 28% reduced risk of breast cancer-related deaths compared to G-CSF_low_ tumours, independent of the standard clinicopathological factors including age, tumour size, grade, lymphovascular invasion, and axillary lymph node metastasis (HR 0.72, CI 0.55–0.93; *p* = 0.01) (Figure 3). Within non-luminal subgroups, G-CSF_high_ expression was significantly associated with better BCSS in HER2 subtype (*n* = 196); triple-negative subgroup (*n* = 469) and basal (*n* = 274) subgroups of non-luminal breast cancers did not generate hazard ratios that remained significant in multivariable analysis (data not shown).

### 3.2. Correlation and Prognostic Significance of G-CSF and CAIX with CD163+ M2 Macrophages and Immune Biomarkers

Non-luminal breast cancers contain regions of hypoxia and acidosis [47,48]. Preclinical studies have shown that G-CSF derived from the hypoxic tumour cells is crucial for mobilization of myeloid-derived suppressor cells to visceral organs, thus increasing the metastatic potential of breast cancer cells [22]. In line with this preclinical evidence, we observed a significant positive correlation between expression of G-CSF and CAIX (Table 1). Amongst non-luminal cases with a G-CSF_high_ phenotype, significantly adverse BCSS (HR 1.74, CI 1.18–2.56; *p* = 0.004) was observed in CAIX-expressing tumours (Figure 4A). Similar results were observed for OS and RFS (Figure 4B,C), indicating that the expression of hypoxia-induced CAIX adversely impacted the prognosis of patients with non-luminal breast cancer with high tumoural expression of G-CSF. G-CSF_high_ phenotype was associated with significantly better BCSS in CAIX-negative non-luminal cases expressing moderate to high CD163+M2 macrophages; however, no significant difference in survival was observed when analysis was restricted to CAIX-positive cases (Appendix A).

Tumour-associated macrophage infiltration in invasive breast cancer is associated with adverse prognostic parameters. In agreement with previous reports [49], we also found that the presence of moderate to dense infiltrates of CD163+ M2 macrophages was associated with poor prognosis (Appendix A).

Hypoxic and acidic tumour microenvironments are host to immunosuppressive cells including CD163+ M2 macrophages. Moreover, it has been shown that lactic acidosis induces a phenotypic switch from M1 to M2 macrophages, which supports tumour cell proliferation [33]. Considering the association of CAIX with hypoxia and acidosis, we performed an exploratory analysis to assess the prognostic significance of combinatorial expression of CAIX, CD163+ M2 macrophages, and G-CSF in non-luminal cases. We found that compared to the expression of all three biomarkers, the CAIX_negative_/G-CSF_high_ phenotype was associated with significantly improved relapse-free survival (HR 0.59, CI 0.4–0.86; *p* = 0.007), with a similar favourable trend observed for breast cancer-specific and overall survival (Figure 5), suggesting that hypoxia and acidosis are influential to the prognostic association of M2 macrophages in non-luminal breast cancer.

Exhausted T cells either overexpress inhibitory receptors (including programmed death receptor-1 (PD-1), lymphocyte activation gene-3 (LAG-3), and T-cell immunoglobulin domain and mucin domain-3 (TIM3)) or downregulate normal T-cell responses by increasing FOXP3 regulatory T cells [50]. To address this relationship in our cohort, we evaluated the correlation of CAIX and G-CSF with the above immune biomarkers, which have been previously assessed in this tissue microarray series [40,41,42,43]. We found that tumours with G-CSF_high_ and CAIX+ phenotypes independently displayed highly significant positive correlations with the presence of intratumoural lymphocytes expressing CD8, PD-1, FOXP3, TIM3, and LAG3, with carcinoma cells expressing PD-L1, and with CD163+ M2 macrophages (Table 2).

Considering the immune-modulating role of G-CSF, we performed a multivariable analysis to investigate the individual prognostic significance of immune checkpoint biomarkers in G-CSF-expressing non-luminal cases. We found that intratumoural lymphocytes expressing CD8, PD-1, FOXP3, TIM3, and LAG3, as well as PD-L1-expressing carcinoma cells, were each associated with better survival, independent from standard clinicopathologic factors (Table 3). Similarly, most immune biomarkers maintained prognostic significance when the analysis was restricted to CAIX-expressing non-luminal cases (Appendix A).

Taken together, our results demonstrate that the prognostic value of G-CSF in non-luminal breast cancers is influenced by tumour microenvironmental features associated with CAIX positivity.

## 4. Discussion

We herein report the prognostic significance of G-CSF expression on breast carcinoma cells in a large population-based cohort of stage I–III invasive breast cancers. The most salient findings of our study were (a) a survival advantage associated with high expression of G-CSF in non-luminal subtypes of breast cancer; (b) a novel association between G-CSF, CAIX, and markers of immune exhaustion on tumour-infiltrating lymphocytes, which is consistent with the presence of an immune-suppressive hypoxic and acidic tumour microenvironment; (c) the identification of a patient subset among non-luminal breast cancers with high G-CSF and CAIX expression that is associated with worse survival compared to those that did not express CAIX; and (d) a positive correlation between tumour-infiltrating lymphocytes expressing immune checkpoint biomarkers and carcinoma cells expressing G-CSF, such that their concurrent presence is associated with a survival advantage in non-luminal immune-enriched subtypes.

The G-CSF/G-CSFR signalling axis has been implicated in stimulating neoangiogenesis, tumour cell proliferation, enhanced migratory and metastatic potential, and expansion of the cancer stem cell pool in preclinical models of several solid organ malignancies [51]. Additionally, there is an abundance of evidence to support that bidirectional signalling cues between G-CSF-mobilized immune cells and tumour-infiltrating lymphocytes can either inhibit or stimulate tumour progression by modulating both the innate and adaptive immune responses. For instance, using transgenic mouse models, it has been shown that mammary tumour-derived cytokines upregulate G-CSF production and facilitate tumour dissemination through expansion and reprogramming of neutrophils, which in turn restrain the effector functions of CD8+ cytotoxic T cells through production of inducible nitric oxide synthase [3]. Likewise, in co-culture experiments, G-CSF secretion by MDA-MB-231 breast cancer cells induces a phenotypic switch in peripherally derived monocytes toward immunosuppressive TAMs with enhanced migratory and metastatic potential, which could be abrogated by anti-G-CSF antibodies [32]. In another independent study, using 4T1 murine breast cancer cells, tumour-derived G-CSF was instrumental in promoting hematopoietic stem cell differentiation toward myeloid lineages, with expansion and activation of CD11b+ Ly6G+ neutrophils, which induce T-cell suppression through the production of reactive oxygen species [52]. Our data on clinical specimens confirms that G-CSF is associated with immune infiltration and supports an immunomodulatory role of G-CSF in the breast tumour microenvironment.

It is well documented that tumour hypoxia and ensuing acidification of the microenvironment promote immunological escape and resistance to immunotherapy by several mechanisms [53,54]. Firstly, hypoxic stress creates nutrient competition between the tumour cells and T lymphocytes such that the resulting nutrient deficit profoundly suppresses the T-cell effector functions leading to a state of hyporesponsiveness even in highly antigenic tumours [55]. Secondly, hypoxia-induced chemokine (CCL28) production [56] and *FOXP3* transcriptional upregulation enriches for FOXP3+ T regulatory cells, which play a crucial role in self-tolerance [57]. Thirdly, hypoxic stress, via induction of hypoxia-inducible factor 1-α, increases the expression of exhaustion markers such as PD-L1 on the tumour cells, which bind to the PD-1 receptor on the surface of T cells, causing effector cell dysfunction and apoptosis [58]. Moreover, hypoxia-induced CAIX expression has been found in tumours resistant to anti-PD-1 therapy [59]. Consistent with this, we have previously demonstrated, in preclinical models of basal-like breast cancer and malignant melanoma, that CAIX expression is associated with an altered anti-tumour immune response and that its inhibition enhances the efficacy of immune checkpoint blockade [23]. Our data here reinforces our previous findings by identifying that CAIX expression at the protein level is associated with the expression of immune checkpoint molecules by the carcinoma cells and by infiltrating immune cells. Furthermore, the hypoxia and acidosis commonly associated with CAIX expression, and the immune modulation associated with these features, may be critical factors driving the prognostic dichotomization of G-CSF_high_ and CD163+ M2 macrophage-infiltrated non-luminal breast cancers.

The identification of a soluble marker, G-CSF, whose expression is associated with the expression of CAIX in a clinical cohort, is an important finding. Our preclinical findings linked CAIX and G-CSF through the NF-κB signalling pathway [22]. Non-luminal breast cancers are known to rely on an active NF-κB signalling pathway; it is therefore plausible that this relationship exists in clinical samples to a certain degree [60,61]. Furthermore, several studies have described a role of G-CSF in tumour progression using the same hypoxic, CAIX-expressing 4T1 model [62,63,64] or additional models containing significant levels of hypoxia such as Lewis lung carcinoma [65] and MMTV-PyMT [66], suggesting a critical role of hypoxia in regulating G-CSF biology in these models. Importantly, cytokine networks in the hypoxic tumour microenvironment, including those orchestrated by TGF-β, may cooperate with G-CSF to potentially influence neutrophil polarization [67]. However, further experimental work is needed to fully elucidate these mechanisms [68].

Differences between the prognostic significance of G-CSF in our study and those previously reported in other tumour types may be attributed to fundamental differences in the underlying tumour biology. For example, inactivation of tumour suppressor pVHL is observed in up to 80% of clear cell renal cell carcinomas, yet this is rare in breast carcinomas [69,70]. The association of G-CSF with poor prognosis in these tumours [14] may be related to a predominant acidotic microenvironment due to constitutive activation of hypoxia-inducible factor 1-α and upregulation of *CA9,* which encodes CAIX [69]. Similarly, tumour hypoxia and CAIX expression are critical drivers [71,72] of the aggressive biology in patients diagnosed with cervical cancer and are likely to influence the prognostic capacity of G-CSF [15]. Hence, it is plausible that in the context of low hypoxic stress and consequently low CAIX, high tumoural G-CSF confers a survival advantage in breast cancer, as observed in our non-luminal cohort (Figure 5).

To date, limited studies have investigated the significance of G-CSF protein expression in breast cancer. Considering the strong association with neutrophilic mobilization and recruitment in carcinomas, serum G-CSF has been evaluated as a surrogate biomarker for prognostication. In this context, a few studies have shown a higher plasma level of G-CSF in breast cancer patients compared to healthy controls and post-surgical wound-healing fluids [73,74,75]. Since these studies were limited to smaller cohorts, meaningful associations with breast cancer subtypes or prognosis could not be addressed. To the best of our knowledge, only a single study has examined the immunohistochemical expression of G-CSF in a reasonably large breast cancer cohort (548 cases), reporting a significant positive correlation between G-CSF expression and CD163+ tumour-associated macrophages [32]. In a further subgroup analysis, it was shown that triple-negative breast cancer cases (*n* = 127) with high G-CSF expression were associated with poor overall survival [32]. Though the prognostic significance of G-CSF in this study is opposite of our observation, these discrepancies may be attributed to the small sample size of the earlier study, differences in the endpoints for survival analysis, the IHC protocols, and perhaps the level of CAIX expression in their cohort.

In a recent study, using publicly available data sets, *CSF3R* was identified as one of the differentially expressed genes in a subset of immune-rich ER+ breast cancers [76]. Although expression analysis of G-CSFR was not included in our study, we performed an exploratory analysis to address whether G-CSF expression contributed to prognostication in a subset of ER+ breast cancers with ≥10% TIL count (*n* = 284), and we observed no significant difference in survival between G-CSF_high_ and G-CSF_low_ tumours (data not shown).

Our study has major strengths, including the use of analytically validated antibodies, predefined scoring criteria (CD163 and CAIX), and the use of an independent cohort (cohort I) to confirm the performance of G-CSF as a biomarker on breast cancer tissue microarrays. However, there are some important limitations worthy of mention. First, we had hoped to see significant findings in cohort I (which, being a training set, is at risk for overfitting) that could guide prespecified formal hypotheses testing using cohort II as an independent validation set. However, the results on the training set were not significant for our primary endpoint in univariable analysis. Instead, we were able to use the findings from cohort I to lock down the staining methodology, interpretation, and cutpoints, but our observations using the more powered cohort II with longer-term follow-up will need independent validation. Secondly, we did not evaluate the presence of tumour-associated neutrophils or myeloid-derived suppressor cells, which are the two major immune cell types mobilized and recruited to the tumour site by G-CSF. Instead, we took into consideration the expression of immune checkpoints on tumour-infiltrating lymphocytes as a related biomarker, hypothesizing that G-CSF mobilizes immune infiltrates and promotes an immune exhaustion phenotype on tumour-infiltrating lymphocytes. Thirdly, we limited our analysis to G-CSF expression by the carcinoma cells, and these results do not take into account the expression of G-CSF on immune cells, such as macrophages, endothelial cells, or other stromal cells that are part of the tumour immune microenvironment. The use of tissue microarrays may be considered a potential limitation of our study. While there are more than a few studies showing considerable agreement between cores derived from source tumour blocks and the full-face sections [77,78], this may not be the case for biomarkers that show high intratumoural heterogeneity and/or predominant expression in the tumour’s microenvironment. Furthermore, this study preceded the time when HER2-targeted therapies and taxanes were routinely used in adjuvant settings. This may limit the extrapolation of prognostic associations to patients receiving the current standard of care therapeutics and merits validation on a more contemporary or prospective series.

## 5. Conclusions

In conclusion, our results demonstrate the prognostic significance of G-CSF in invasive breast cancer, whereby high expression serves as an indicator of better survival in aggressive non-luminal subtypes of breast cancer in the absence of CAIX. Our identified associations between CAIX, G-CSF, and immune biomarkers provide a rationale for additional prospective investigations to understand the underlying mechanisms and their role as potential biomarkers for predicting responses to immune checkpoint inhibitor therapy.

## Figures and Tables

**Figure 1 cancers-13-01022-f001:**
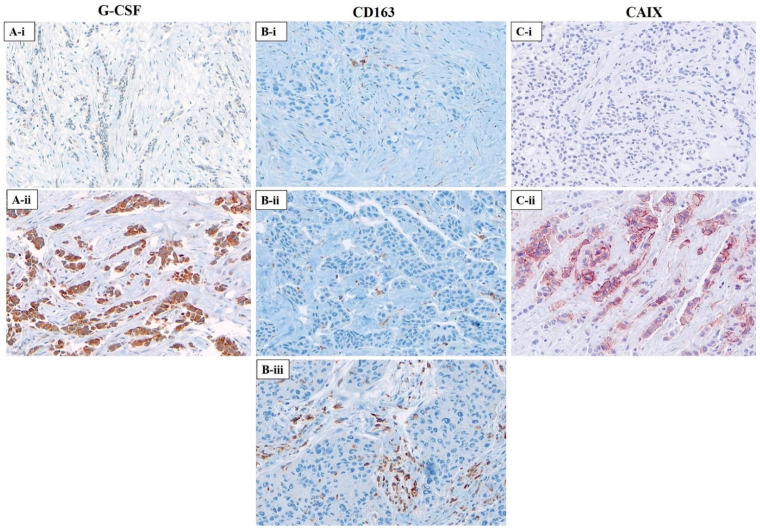
Representative photomicrographs for immunohistochemical staining of G-CSF, CD163, and carbonic anhydrase IX (CAIX) in breast carcinoma tissue microarray (cohort II) (Images acquired at 200X). (**A**) Cytoplasmic expression of G-CSF on breast carcinoma cells; (**A-i**) low (≤1); (**A-ii**) high (>1); (**B**) Membranous or cytoplasmic expression of CD163 on tumour-associated macrophages; (**B-i**) sparse infiltrates, ≤5 stained macrophages; (**B-ii**) moderate infiltrates, >5 but ≤25 positively stained macrophages; (**B-iii**) dense infiltrates, >25 positively stained macrophages; (**C**) Membranous expression of CAIX on breast carcinoma cells; (**C-i**) negative; (**C-ii**) positive.

**Figure 2 cancers-13-01022-f002:**
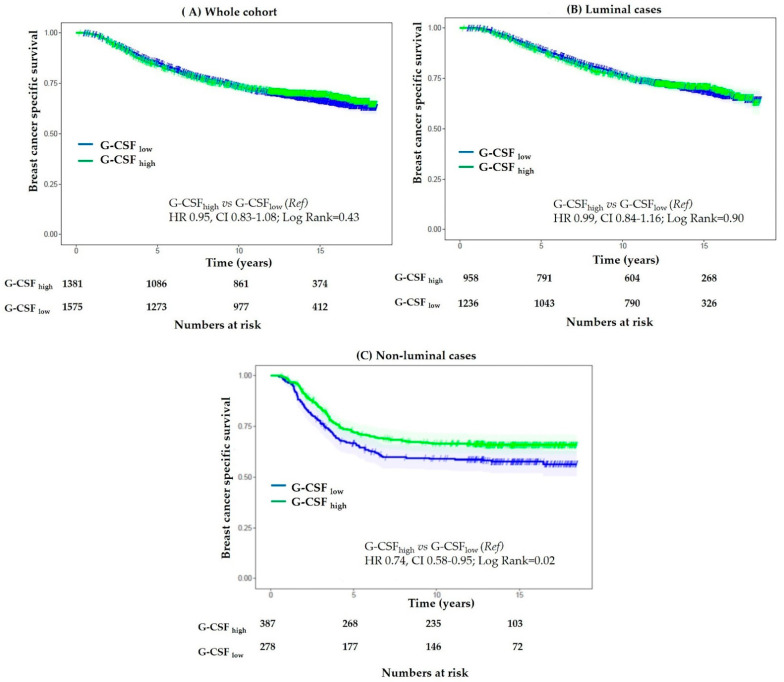
Cohort II: Kaplan–Meier curves: Association of G-CSF expression with breast cancer-specific survival (BCSS). No significant difference in BCSS was observed in the whole cohort (**A**) and cases with luminal subtype (**B**). An improved BCSS was observed in cases with non-luminal subtype (**C**).

**Figure 3 cancers-13-01022-f003:**
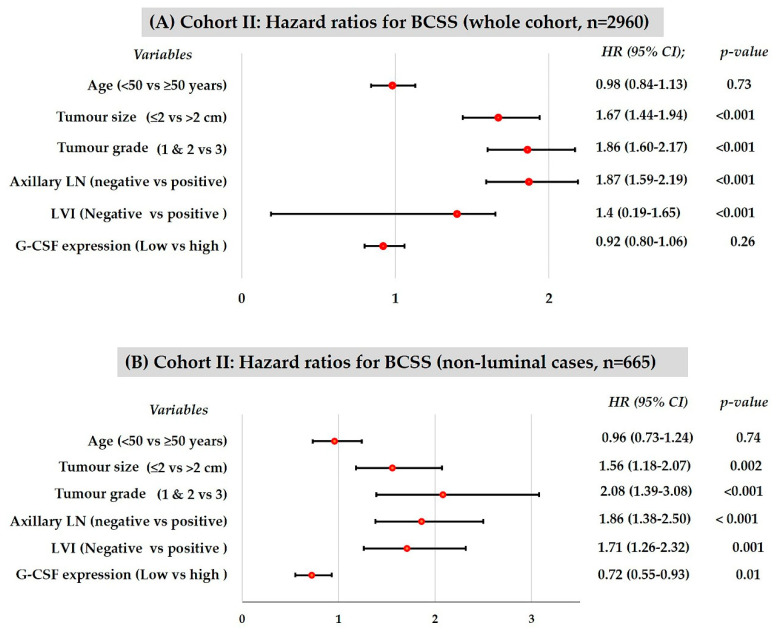
Cohort II: Forest plots based on the results of multivariable analysis for risk factors associated with breast cancer-specific survival in whole series (**A**) and in non-luminal cases (**B**).

**Figure 4 cancers-13-01022-f004:**
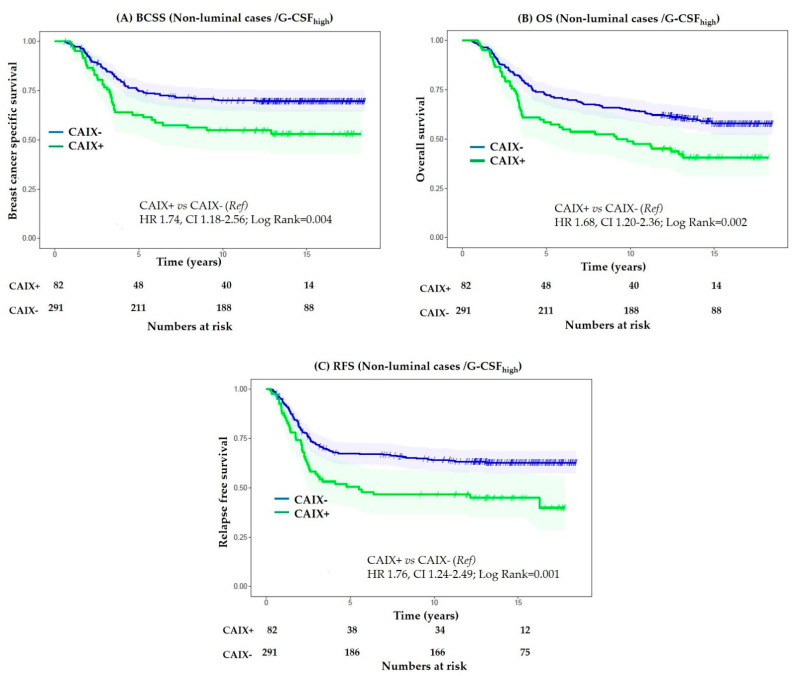
Kaplan–Meier curves for association of CAIX expression in non-luminal cases with G-CSF_high_ phenotype. Expression of CAIX is associated with adverse breast cancer-specific survival (**A**), overall survival (**B**), and relapse-free survival (**C**).

**Figure 5 cancers-13-01022-f005:**
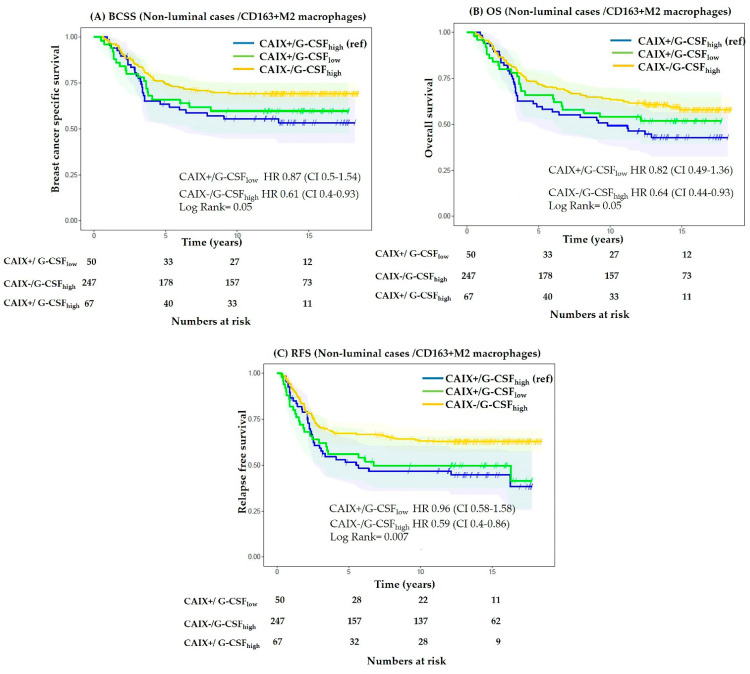
Kaplan–Meier curves for association of CAIX and G-CSF in CD163+ non-luminal cases. Breast cancer-specific survival (**A**), overall survival (**B**), relapse-free survival (**C**). CAIX_negative_/G-CSF_high_ phenotype is associated with significantly better relapse-free survival compared to CAIX_positive_/G-CSF_low_ phenotype (**C**).

**Table 1 cancers-13-01022-t001:** Correlation of G-CSF expression with clinicopathological features and other biomarkers.

Clinicopathological Variables	G-CSF Expression		*p*-Value
	Low (≤1)	High (>1)	
**Age at diagnosis**			0.02
<50	452 (28.7)	451 (32.6)	
≥50	1125 (71.3)	932 (67.4)	
**Menstrual status**			0.02
Premenopausal	434 (28.1)	434 (32)	
Postmenopausal	1111 (71.9)	922 (68)	
**Tumour size (cm)**			0.06
≤2	784 (50)	736 (53.4)	
>2	785 (50)	641 (46.6)	
**Tumour grade**			0.92
1 & 2	683 (45.5)	606 (45.3)	
3	818 (54.5)	731 (54.7)	
**Axillary lymph node status**			0.91
Negative	888 (56.5)	783 (56.7)	
Positive	684 (43.5)	598 (43.3)	
**Lymphovascular invasion**			0.31
Negative	837 (55.5)	714 (53.6)	
Positive	671 (44.5)	618 (46.4)	
**ER expression**			<0.001 *
Negative	359 (22.8)	449 (32.5)	
Positive	1213 (77.2)	932 (67.5)	
**PR expression**			0.57
<1%	699 (47.1)	642 (48.2)	
≥1%	785 (52.9)	691 (51.8)	
**HER2 overexpression/amplification**			<0.001 *
Negative	1392 (90.5)	1125 (82.6)	
Positive	146 (9.5)	237 (17.4)	
**CK5/6 expression**			<0.001 *
Negative	1301 (93.2)	1112 (89.2)	
Positive	95 (6.8)	135 (10.8)	
**EGFR expression**			<0.001 *
Negative	1281 (90.3)	1059 (83.2)	
Positive	137 (9.7)	214 (16.8)	
**Ki-67 proliferation index**			0.03
<14%	800 (56.2)	672 (52)	
≥14%	624 (43.8)	621 (48)	
**CAIX expression**			<0.001 *
Negative	1271 (86.5)	1066 (81)	
Positive	199 (13.5)	250 (19)	
**CD163+ TAMs**			<0.001 *
Sparse (≤5)	567 (41.4)	431 (34)	
Moderate (>5 ≤ 25)	443 (32.4)	417 (32.9)	
Dense (>25)	359 (26.2)	419 (33.1)	
**Treatment**			0.001 *
No systemic therapy	659 (41.8)	594 (43)	
Tamoxifen only; no chemotherapy	536 (34)	407 (29.4)	
Chemotherapy only; no hormonal therapy	260 (16.5)	292 (21.1)	
Chemotherapy + Tamoxifen	110 (7)	88 (6.4)	
Others	12 (0.8)	2 (0.1)	
**Breast cancer subtypes (IHC-based)**			<0.001 *
Luminal-NOS	96 (6.1)	38 (2.7)	
Luminal A	680 (43.1)	523 (37.8)	
Luminal B	380 (24.1)	302 (21.8)	
Luminal/HER2+	82 (5.2)	97 (7)	
HER2	61 (3.9)	135 (9.8)	
Basal	106 (6.7)	168 (12.1)	
Unassignable	61 (3.9)	36 (2.6)	
Additional Basal if by TNP	111 (7)	84 (6.1)	

* Denotes differences between low and high G-CSF groups that are significant at the Bonferroni-corrected *p*-value of <0.0031 (=0.05/16). G-CSF, granulocyte colony-stimulating factor; ER, estrogen receptor; PR, progesterone receptor; EGFR, epidermal growth factor receptor; HER2; human epidermal growth factor receptor 2; CK, cytokeratin; CAIX, carbonic anhydrase IX; TAMs, tumour-associated macrophages; TNP, triple-negative phenotype (ER-, PR- and HER2-); NOS, not otherwise specified; IHC, immunohistochemistry.

**Table 2 cancers-13-01022-t002:** Correlation of immune biomarkers with G-CSF and CAIX expression (whole cohort)**.**

Variables	G-CSF Expression		*p*-Value	CAIX Expression		*p*-Value
	Low (≤1)	High (>1)		Negative	Positive	
**H & E sTIL count (%)**			<0.001			<0.001
<10	1244 (85.7)	1029 (79)		2539 (83)	170 (71.4)	
≥10	207 (14.3)	274 (21)		520 (17)	68 (28.6)	
**CD8 iTIL count**			<0.001			<0.001
<1	1058 (70)	830 (62)		2001 (67.9)	128 (56.1)	
≥1	454 (30)	508 (38)		945 (32.1)	100 (43.9)	
**PD-1 iTIL count**			<0.001			<0.001
<1	1337 (94.1)	1161 (88.2)		2346 (92.4)	146 (75.6)	
≥1	84 (5.9)	155 (11.8)		192 (7.6)	47 (24.4)	
**PDL1+ tumour cells (%)**			<0.001			<0.001
0	1332 (94.2)	1166 (88.9)		2367 (92.4)	150 (79.8)	
≥1	82 (5.8)	146 (11.1)		194 (7.6)	38 (20.2)	
**FOXP3 iTIL count**			<0.001			<0.001
<2	1087 (71.9)	831 (61.6)		1951 (68.4)	133 (59.9)	
≥2	425 (28.1)	518 (38.4)		900 (31.6)	89 (40.1)	
**TIM3 iTIL count**			<0.001			<0.001
<1	1360 (90)	1182 (87.4)		2453 (89.8)	165 (79.3)	
≥1	151 (10)	171 (12.6)		279 (10.2)	43 (20.7)	
**LAG3 iTIL count**			<0.001			<0.001
<1	1297 (91.1)	1136 (85.9)		2283 (89.4)	148 (75.5)	
≥1	126 (8.9)	186 (14.1)		271 (10.6)	48 (24.5)	
**CD163+ TAMs**			<0.001			<0.001
Sparse (≤5)	567 (41.4)	431 (34)		1048 (37.8)	44 (21.1)	
Moderate (>5 ≤ 25)	443 (32.4)	417 (32.9)		907 (32.7)	64 (30.6)	
Dense (>25)	359 (26.2)	419 (33.1)		820 (29.5)	101 (48.3)	

H & E, hematoxylin and eosin-stained; sTIL, stromal tumour-infiltrating lymphocytes; iTIL, intratumoural tumour-infiltrating lymphocytes; CD, cluster of differentiation; PD-1, programmed cell death protein-1; PD-L1, programmed death ligand-1; FOXP3, forkhead box protein P3; TIM3, T-cell immunoglobulin and mucin domain-containing molecule 3; LAG-3, lymphocyte activation gene 3; TAMs, tumour-associated macrophages.

**Table 3 cancers-13-01022-t003:** Multivariable analysis for prognostic significance of immune biomarkers in non-luminal cases with G-CSF_high_ phenotype.

Covariates	BCSS	
	Non-Luminal Cases/G-CSF_high_	
	HR (95% CI)	*p*-Value
**Age at diagnosis**		0.16
<50	1	
≥50	0.76 (0.52–1.12)	
**Tumour size (cm)**		0.004
≤2	1	
>2	1.82 (1.22–2.71)	
**Tumour grade**		0.04
1 & 2	1	
3	1.80 (1.02–3.17)	
**Axillary lymph node status**		0.04
Negative	1	
Positive	1.59 (1.03–2.46)	
**LVI**		0.58
Negative	1	
Positive	1.14 (0.73–1.77)	
**H & E stromal TILs (%)**		0.001
<10	1	
≥10	0.48 (0.31–0.72)	
**Age at diagnosis**		0.06
<50	1	
≥50	0.69 (0.48–1.01)	
**Tumour size (cm)**		0.01
≤2	1	
>2	1.72 (1.17–2.55)	
**Tumour grade**		0.02
1 & 2	1	
3	2.06 (1.15–3.70)	
**Axillary lymph node status**		0.10
Negative	1	
Positive	1.43 (0.93–2.19)	
**LVI**		0.07
Negative	1	
Positive	1.49 (0.97–2.31)	
**CD8 iTIL count**		0.01
<1	1	
≥1	0.59 (0.40–0.90)	
**Age at diagnosis**		0.17
<50	1	
≥50	0.77 (0.52–1.12)	
**Tumour size (cm)**		0.05
≤2	1	
>2	1.50 (1.01–2.23)	
**Tumour grade**		0.02
1 & 2	1	
3	2.03 (1.13–3.64)	
**Axillary lymph node status**		0.05
Negative	1	
Positive	1.55 (0.99–2.43)	
**LVI**		0.18
Negative	1	
Positive	1.37 (0.87–2.16)	
**PD1 iTIL count**		<0.001
<1	1	
≥1	0.36 (0.20–0.63)	
**Age at diagnosis**		0.08
<50	1	
≥50	0.71 (0.49–1.04)	
**Tumour size (cm)**		0.02
≤2	1	
>2	1.59 (1.08–2.33)	
**Tumour grade**		0.01
1 & 2	1	
3	2.13 (1.19–3.83)	
**Axillary lymph node status**		0.02
Negative	1	
Positive	1.64 (1.08–2.50)	
**LVI**		0.18
Negative	1	
Positive	1.35 (0.88–2.07)	
**FOXP3 iTIL count**		0.002
<2	1	
≥2	0.55 (0.37–0.80)	
**Age at diagnosis**		0.12
<50	1	
≥50	0.75 (0.51–1.08)	
**Tumour size (cm)**		0.01
≤2	1	
>2	1.67 (1.14–2.46)	
**Tumour grade**		0.03
1 & 2	1	
3	1.85 (1.05–3.24)	
**Axillary lymph node status**		0.04
Negative	1	
Positive	1.57 (1.03–2.40)	
**LVI**		0.2
Negative	1	
Positive	1.33 (0.86–2.05)	
**TIM3 iTIL count**		0.01
<1	1	
≥1	0.48 (0.28–0.84)	
**Age at diagnosis**		0.16
<50	1	
≥50	0.76 (0.52–1.12)	
**Tumour size (cm)**		0.01
≤2	1	
>2	1.69 (1.14–2.51)	
**Tumour grade**		0.01
1 & 2	1	
3	2.09 (1.16–3.75)	
**Axillary lymph node status**		0.04
Negative	1	
Positive	1.58 (1.02–2.44)	
**LVI**		0.23
Negative	1	
Positive	1.31 (0.84–2.04)	
**LAG3 iTIL count**		0.001
<1	1	
≥1	0.45 (0.28–0.73)	
**Age at diagnosis**		0.05
<50	1	
≥50	0.67 (0.46–0.99)	
**Tumour size (cm)**		0.01
≤2	1	
>2	1.68 (1.13–2.50)	
**Tumour grade**		0.03
1 & 2	1	
3	1.92 (1.07–3.45)	
**Axillary lymph node status**		0.03
Negative	1	
Positive	1.62 (1.04–2.51)	
**LVI**		0.27
Negative	1	
Positive	1.29 (0.82–2.01)	
**PD-L1+ tumour cells (%)**		0.01
0	1	
≥1	0.46 (0.26–0.83)	
**Age at diagnosis**	0.79 (0.54–1.15)	0.21
<50		
≥50		
**Tumour size (cm)**		0.01
≤2	1	
>2	1.17 (1.16–2.53)	
**Tumour grade**		0.04
1 & 2	1	
3	1.85 (1.03–3.30)	
**Axillary lymph node status**		0.11
Negative	1	
Positive	1.42 (0.92–2.19)	
**LVI**		0.15
Negative	1	
Positive	1.39 (0.89–2.16)	
**CD163+ M2 macrophages**		
Sparse	1	
Moderate	1.12 (0.63–2.0)	0.70
Dense	0.78 (0.44–1.35)	0.37

## Data Availability

The data presented in this study will be available on request from the corresponding author. Images from the tissue microarray slides are available for public access via the website of the Genetic Pathology Evaluation Center (http://www.gpec.ubc.ca/gcsf accessed on 1 January 2021).

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
