# Peer review of "Granulocyte Colony Stimulating Factor Expression in Breast Cancer and Its Association with Carbonic Anhydrase IX and Immune Checkpoints"

_cancers, 2021, doi:10.3390/cancers13051022_

Round 1

Reviewer 1 Report

In a triple negative cohort, high G-CSF correlated with a better prognosis, despite the fact that high G-CSF correlated with high CAIX, which was shown to be associated with a poorer prognosis.  The important point that emerges from this study is that the predictive value of tumor G-CSF is dependent on the levels of CAIX in the tumor, with high G-CSF only being predictive of good outcome when CAIX levels are low.

Likewise, high G-CSF is positively correlated with high levels of CD163+ macrophages, another marker of poor prognosis. Multivariate analysis of tumors with high G-CSF levels demonstrated that a number of immune cell biomarkers were associated with a better prognosis, despite these elevated checkpoint proteins being an indication of T cell exhaustion.

The preclinical studies mentioned in the Discussion (lines 333-350) and many others that should be referenced here, all point to high G-CSF being associated with poor outcome.  In these preclinical studies, the mice are not usually given chemotherapy.  One possible explanation could be the fact that patients with TNBC are more likely to receive chemotherapy than those with luminal breast cancer.  Chemotherapy is associated with neutropenia, sometimes requiring a pause in drug delivery or supplementation with G-CSF.  Is it possible that the patients whose tumors are already high in G-CSF are more likely to tolerate the chemotherapy without interruption, and thereby achieve a better outcome?  Do the authors have data on the chemotherapy regimens and G-CSF supplementation given to patients in this cohort?

The discrepancy between the preclinical studies and this immunohistological study of patient samples warrants further discussion.  Are all the preclinical studies confounded by high levels of CAIX?

From the data in Table 2, high CAIX also seemed to be associated with increased checkpoint protein levels, but CAIX is a poor prognostic factor.  What is the outcome of a multivariate analysis of immune biomarkers in cancers with high CAIX levels?

The authors identify a group of patients who have both high G-CSF and high CAIX, who have a poorer outcome than those with low CAIX and provide a rational explanation for this in the Discussion, with regard to ensuing hypoxia and acidification.  They mention that this leads to increased expression of exhaustion markers on T cells, such as increased PD-1 and increased tumor expression of PD-L1.  However, high levels of PD-1/PD-L1 are typically indicative of a good response to immune checkpoint blockade, which should lead to improved outcome.  Presumably these patients did not receive checkpoint inhibitors.

In this study, PD-L1 levels on tumor cells are reported.  However, myeloid cells – macrophages and neutrophils - also express PD-L1.  What is the relevance of this source of PD-L1 in triple negative breast cancer prognosis?

G-CSF is known to recruit neutrophils into tumors, but neutrophils have not been assessed in this study.  Why not?

Minor comments:

  1. Table 1: please define TNP.
  2. Please clarify the numbers for different subtypes of cancer, as given in Table 1 and then mentioned in the text below Figure 2. For the K-M analysis in Figure 2C, a total of 665 non-luminal cases are included (387 hi and 278 lo). However, in the text below this figure, a number of 880 non-luminal cases is included for the Cox proportional hazards ratio for Figure 3.  Why is the number different?  Also, the same number (880) is assigned to the Her2+ cases, but from Table 1, these add up to 375 cases.  The numbers given for the triple negative and basal subgroups also do not seem to match those in Table 1.
  3. Can you explain why G-CSF levels correlated significantly with better OS in the total Cohort 1 but not total Cohort 2? Is there something different about Cohort 1 in terms of therapy given to these patients?  Or is it just a numbers game - fewer samples in Cohort 1?  There were very few TN cancers in Cohort 1 so these would be unlikely to skew the result towards a favorable prognosis.
  4. Part of Figure 5 legend is missing.
  5. Discussion, line 383. The sentence beginning “Additionally, these results….” is confusing. First, the statement is not referenced and second, if it is correct that plasma G-CSF levels can be confounded by the presence of other cytokines, why isn’t the same fact true when considering G-CSF levels in tumors?

Author Response

Re: Manuscript ID: Cancers-1088054

Granulocyte colony stimulating factor expression in breast cancer and its association with carbonic anhydrase IX and immune checkpoints

We are grateful to the Reviewer for the constructive feedback on our manuscript. We believe we have been able to adequately address all comments. Please find our point-by-point responses (in blue font) in the appended file. 

Reviewer 2 Report

In the present study the authors evaluated immunohistochemically in breast cancer TMAs the expression of G-CSF and CAIX on cancer cells and CD163 on immune cells. Authors test the expression of the marker in a large cohort of patients, however showed that the G-CSF expression in breast cancer cells is associated with better prognosis in contrast to previous studies as referred to lane 68.

Minor comments

  1. In paragraph starting from lane 225 the authors mention that G-CSF expression is correlated with ER negativity, HER2 positivity, CAIX positivity, HER2 and Basal subtypes, which is not agree with the data on Table1.
  2. You support that G-CSF expression in breast cancer cells is associated with better disease OS and RSF, how you compensate your statement in lane 68 “clinical studies evaluating G-CSF expression on tumor cells …. renal carcinomas”
  3. The conclusion in lane 311 " Taken together, these findings support......non-luminal breast cancer" should be modified as your findings do not show any modulatory role of G-CSF and CAIX but only the expression profile of these markers.

Major comments

  1. In lane 78, you refer that you have already showed, in previous studies, the expression of CAIX in breast cancer and its significance in disease prognosis, so the novelty of the present study is to evaluate the impact of G-CSF expression on disease prognosis. However, your results in Figures 4 and 5 show that CAIX is the prognostic marker and not G-CSF. To show the significant of G-CSF, you could indicate the survival plots also for the cases with low G-CSF expression in association with CAIX (low vs high cases) and CD163.
  1. In Figure 1, it could be better to show the immunohistochemical staining for each marker in the same core specimen supporting your statistical findings and not show just a representative staining of high, moderate, and low expression for each marker.
  2. A major limitation of the present study is the use of tissue microarrays. It is well known that except from inter-tumor microenvironment heterogeneity, of great significance is also the intra-tumor heterogeneity. Lots of studies have shown that a tumor displays a great heterogeneity as it concerns the cancer cells genetic profile, the immune cells infiltration and the extracellular matrix. Choosing two cores from the tissue and neglecting the rest part, you cannot export conclusions about tumor's microenvironment. it could be better if you stain whole tissue specimens and show the differences within the same tissue for each of the marker you study.

Author Response

(The authors gave the same response as above.)

Round 2

Reviewer 1 Report

The authors have satisfactorily addressed the questions that I raised in my first review.

Correct typographical errors in the legend to Table 1 and in Line #412.

Reviewer 2 Report

the authors cover efficiently most of the comments.